# Effect of Orchard Management Factors on Flesh Color of Two Red-Fleshed Apple Clones

**Annika Wellner, Eckhard Grimm \* and Moritz Knoche** 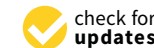

Institute for Horticultural Production Systems, Leibniz-University Hannover, Herrenhäuser Straße 2, 30419 Hannover, Germany

\* Correspondence: eckhard.grimm@obst.uni-hannover.de; Tel.: +49-511-762-8081

**Abstract:** Little is known about factors affecting anthocyanin biosynthesis in red-fleshed apples (*Malus* × *domestica* Borkh.). The objective was to establish the effects of orchard management factors on flesh anthocyanin content of dark-colored (DC) and light-colored (LC) apple clones. Flesh color was assessed by measuring color in the L, a, b mode using a spectrophotometer and predicting the anthocyanin content based on relationships between the absorption of a flesh extract at 530 nm and the L-value determined using a spectrophotometer ($r^2 = 0.99$ \*\*\*). Fruit from the DC clone were red by 86 days after full bloom (DAFB), whereas the LC clone began to color at 136 DAFB. Color intensity in both clones decreased from the top of the tree to the base. Further, the intensity of the flesh color of the DC clone decreased with shading (94% absorption of incident photosynthetic active radiation). Covering a fruit with a UV absorbing film (100% UV absorption) had no effect on flesh color in the DC clone but decreased color in the LC clone. Fruit thinning increased color in DC and LC fruit. There was little change in flesh color during storage. However, the DC clone developed severe flesh browning as storage progressed beyond 30 days. The results demonstrated that light (visible and UV wavelength) stimulated, whereas shade inhibited, anthocyanin biosynthesis in the flesh under orchard conditions.

**Keywords:** *Malus* × *domestica*; anthocyanin; canopy; light interception; parenchyma; crop load; UV light

## 1. Introduction

The red blush of apple fruit skin is an important quality attribute. Recently, some apple genotypes have become available that also have red-flesh. In these genotypes, the red color in the skin extends into the flesh of the fruit. Red-fleshed apples are novel, and this makes them attractive to consumers and so of increased economic value. A number of red-fleshed apple genotypes have now been identified and evaluated [1–3]. Red fleshed genotypes fall into two categories, types 1 and 2. Type 1 genotypes are strongly pigmented and dark-colored. The flesh color forms early during fruit development and the leaves and shoots are also colored. In type 1 genotypes, the transcription factor MdMYB10 is involved in regulating the biosynthesis of anthocyanins. Type 2 genotypes are less color intensive. Coloration begins late in development, and leaves and shoots are without red color. In type 2 genotypes the transcription factor MdMYB110a is involved in the regulation of biosynthesis of anthocyanin [4–6].

The pigments responsible for the red flesh are anthocyanins. In general, these may appear in stems, leaves, flowers or fruits. The principle function of coloration is to make a plant or a plant organ attractive to animals which act as vectors, carrying pollen and dispersing seeds. High anthocyanin concentrations in fruit and vegetables are also considered beneficial in the human diet because of their antioxidant [7], anticancer and antiaging properties [8,9].

The concentration of anthocyanin in the flesh and hence color intensity depends not only on the genotype but also on environmental variables. For example, the anthocyanin content of a type 1 genotype varied by 14% depending on the site of production [10]. In addition, both light and temperature affected anthocyanin biosynthesis in apple, and hence the skin and flesh color; for review see [11,12]. Little is known about the effect of orchard management factors during cultivation on color development in apple flesh.

The objective of our study was to establish the effects of orchard management factors on anthocyanin content of a dark-colored (DC, type 1) and a light-colored (LC, type 2) apple clone. We focused on factors related to light exposure because light affects anthocyanin biosynthesis in the skin and may, therefore, also play a role in the flesh. The two apple clones examined here are representative of the small range of DC and LC type apple fruits currently available.

## 2. Materials and Methods

### 2.1. Plant Material

Two red-flesh apple (*Malus* × *domestica* Borkh.) clones were selected that had dark-red (DC) and light-red (LC) flesh (IFORED SAS, Seiches-sur-le-Loire, France). Trees were grafted on M9 rootstocks and grown at the Obstbauzentrum Jork, Jork, Germany (53.5 N, 9.7 E) according to current practices of integrated fruit production. Fruits were harvested during the season beginning at 86 days after full bloom (DAFB) and were processed immediately or stored at 2 ± 1 °C pending analysis.

### 2.2. Quantifying Redness of Cross-Sections

For rapid quantification of the redness of the flesh, a procedure based on image analysis of digital photographs was established. Flesh slices (3–5 mm thick) were excised from the equatorial plane of DC and LC fruits. Equatorial slices were photographed using a Canon EOS 550 D camera equipped with Macro lens EF-S 18–55 mm (Canon, Tokyo, Japan) mounted on a photo stand (RB 5000 DL, No. 5556 + Prolite 5000, No. 2190; Kaiser Phototechnik, Buchen, Germany) equipped with three fluorescent tubes (Osram TC-L, code 954, 5400 K), two above (36 W each) and one (15 W) in a lightbox underneath the specimen. Images were taken together with the RHS color standards (Charts No. 39, 43, 45, 46; Royal Horticultural Society, London, UK) and the ColorChecker (X-Rite Pantone; Grand Rapids, MI, USA) as shown in Figure 1a. Images were processed using image analysis software (cellSens Dimension 1.7.1; Olympus, Hamburg, Germany) and the following procedure. First, the white standard was calibrated. Second, the cross-section of the flesh was partitioned into four zones of different color intensity. The limits for the different intensities were set according to the RHS color charts (No. 39, 43, 45, 46), as shown by false colors in Figure 1b. As for any one of the color charts the L-, a- and b-coordinates were pre-determined (CM-2600d; Konica Minolta, Osaka, Japan) [13], a mean L-, a mean a- or a mean b-value could be calculated that were weighted by the respective area of the cross-section.

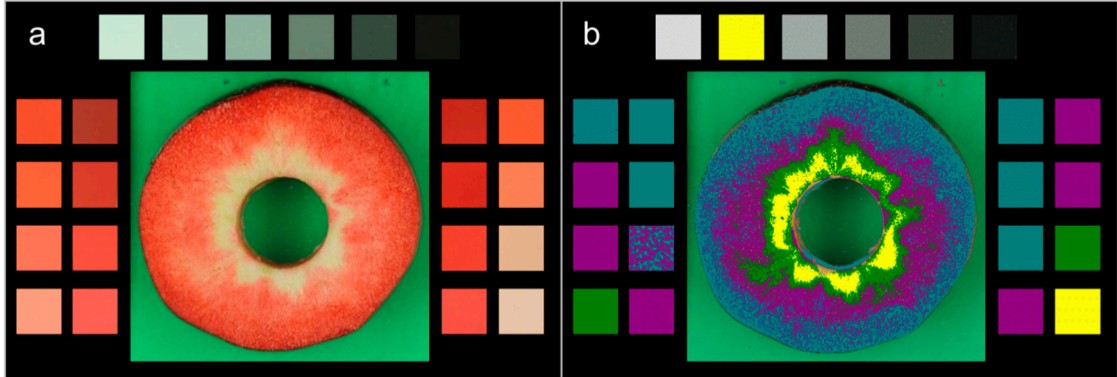

**Figure 1.** (**a**) Image of the equatorial slice of a red-fleshed apple that was taken together with four red color charts (No. 39, 43, 45, 46) of the Royal Horticultural Society (RHS) and a grey scale of ColorChecker. Red color charts and grey scale served as calibration. (**b**) False-color image of the photograph depicted in A. The four color classes were identified based on the L-values of the RHS color charts.

To relate the L-, a- and b-values determined on cross-sections of the fruit to the anthocyanin content of the tissue, a calibration curve was established. Briefly, tissue cylinders (12 mm diameter × 3–5 mm long) were excised from the equatorial plane of DC and LC fruits. Cylinders were selected for homogeneity of color ranging from white to dark red. The color of cylinders was quantified using a spectrophotometer (CM-26001; Konica Minolta). Thereafter, cylinders were extracted using 1% HCl in methanol (8 mL per cylinder) at room temperature overnight (>12 h). The absorption of the extract was quantified spectrophotometrically at 530 nm (Specord 210; Analytik Jena, Jena, Germany). An exponential regression model was fitted between absorption of the extract at 530 nm and the L- and the a-value. Statistical analysis revealed highly-significant relationships between the absorption at 530 nm and the L- or the a-value determined on the same tissue cylinder. The regression equation allowed conversion of any L-value (or a-value) of the three-dimensional color space determined on a fruit cross-section, to an anthocyanin absorption value determined on a flesh extract.

### 2.3. Experiments

A developmental time course was established by sampling the DC clone from 86 DAFB and the LC clone from 96 DAFB to maturity.

Flesh color was recorded as a function of the position of the fruit in the canopy and the exposure of the fruit relative to the sun. The three positions in the canopy were: in the upper third of the crown ('top'), in the middle third ('middle') and in the lower third ('bottom'). Exposure relative to the sun was also varied by selecting fruit from the north- or south-exposed flanks of the row.

Flesh color was recorded as a function of shading. Shading was achieved by wrapping individual branches in the middle of the tree with a hail net. Branches without net served as control. The light intensity in the shade treatment was only about 6% (LI-COR Quantum/Radiometer/Photometer LI250; Heinz Walz GmbH Mess- und Regelungstechnik, Effeltrich, Germany).

Flesh color was recorded as a function of UV exposure. Fruits in the middle of the crown were covered with a UV-absorbing film (UV Schutzfolie Klar UV 3 SR; Özek and Wagner, Cologne, Germany). This film blocked all UV radiation as determined using a UV meter (sensor: type 2.5; Indium Sensor GmbH, Neuenhagen, Germany; data logger: Almemo® 2590-4AS; Ahlborn Mess- und Regelungstechnik GmbH, Holzkirchen, Germany). Fruit in the same position but without film served as control.

Flesh color was recorded as a function of three levels of fruit thinning: no thinning, medium thinning and heavy thinning. Thinning was done by hand after terminal bud set. The mean number of fruits was 200, 120 and 50 per DC and 120, 80 and 30 per LC tree for the light, medium and heavy

thinning treatments, respectively. The mean trunk cross-sectional area (0.6 m above graft union) was 13.37 cm$^2$ and 13.74 cm$^2$ for DC and LC trees, respectively.

The effect of storage duration on flesh color was quantified during 98 d of cold storage at 4 °C.

## 2.4. Data Analysis

All experiments were carried out with a minimum of 20 fruit sampled from a minimum of four trees. Unless individual fruit data are shown, data are means ± standard errors. Where error bars are not shown, they were smaller than the plotted symbols. Data were analyzed by analysis of variance (Proc GLM) and regression analysis (Proc REG) using the statistical software package SAS (version 9.1.3; SAS Institute, Cary, NC, USA). Means were compared using Tukey's Studentized Range (HSD) Test. The significances of coefficients of determination (r$^2$) at $p \leq 0.001$ are indicated by ***.

## 3. Results

Absorption decreased exponentially as the L-value increased (Figure 2a). The curve had a satisfactory resolution, i.e., changes in absorption resulted in measurable ΔL values over the entire range of L-values. Plotting the natural log-transformed absorption vs. the L-value yielded a linear relationship. For the a-value, absorption increased exponentially as the a-value increased (Figure 2b). However, the curve segment for a-values from 30 to 50 was steep, thus the resolution was inferior to that in Figure 2a. When taking the natural log of the absorption and plotting against the a-value, a linear relationship was obtained between the two (Figure 2b, inset). Plots of absorption vs. the b-value yielded a discontinuous curve with separate data populations for DC and LC (data not shown). The relationship between the absorption of anthocyanins and the L-value (or a-value) determined using a spectrophotometer allowed prediction of flesh color as indexed by anthocyanin content from measurements of the L-value (or a-value) on a fruit cross-section. Plotting the predicted vs. measured anthocyanin absorption for an independent data set yielded the relationships shown in Figure 2c. The absorption predicted from L-values was closely and linearly related to the measured absorption (slope 1.12 ± 0.05, r$^2$ = 0.98 ***). The intercept did not differ significantly from zero and hence, the regression line was forced through the origin. Relationships with the a-value were inferior, due to an overestimation of the predicted absorption (slope 1.35 ± 0.13), systematic deviations for medium a-values (r$^2$ = 0.93 ***).

Developmental increase in fruit size followed similar patterns in both clones (Figure 3a). The red color of the flesh of the DC clone decreased slightly between 86 and 112 DAFB and then remained constant until harvest (Figure 3b). In contrast, in the LC clone, the flesh began to color at 136 DAFB and increased continuously in color intensity until 184 DAFB (Figure 3b). Compared with the LC clone, the DC clone had a markedly higher percentage of dark- and medium-colored fruit (Figure 3c,d). There was no relationship between the color of the skin or that of the outer or inner flesh in DC or LC apples (data not shown).

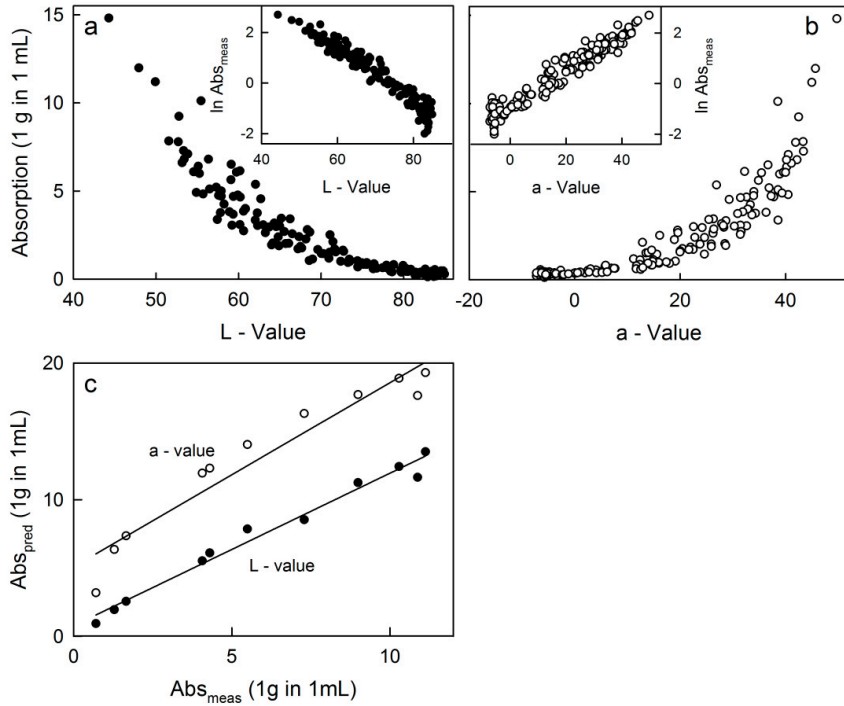

**Figure 2.** Absorption of anthocyanins extracted from the flesh of apples plotted against the color as indexed by the L-value (**a**) or the a-value (**b**). Each data point represents an individual tissue disc. Insets: Data from the main graph replotted, but absorption data ln transformed. (**c**) Relationship between the predicted absorption ($Abs_{pred}$) and the measured absorption ($Abs_{meas}$). Absorption was predicted either from the L-value ($Abs_{pred} = 1.12\ (\pm0.05) \times Abs_{meas} + 0.78\ (\pm0.37)$, $r^2 = 0.98$ ***) or from the a-values ($Abs_{pred} = 1.35\ (\pm0.13) \times Abs_{meas} + 5.08\ (\pm0.89)$, $r^2 = 0.93$ ***). The L- and a-values were determined with a spectrophotometer in the L-, a- and b-modes. For details see text.

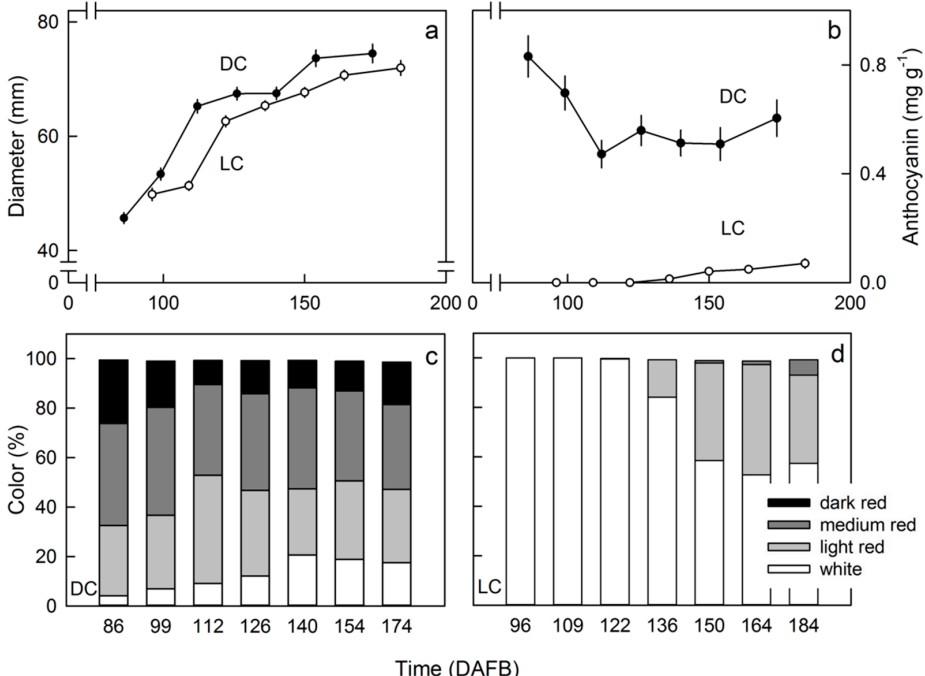

**Figure 3.** Developmental time course of change in diameter (**a**) and flesh color as indexed by the anthocyanin accumulation (**b**) in red flesh apples of a dark-colored (DC) and a light-colored (LC) clone. Distribution of color classes in DC (**c**) and LC clones (**d**).

In both clones, flesh color intensity decreased from top to bottom positions in the tree crown (Figure 4a,b). The color distribution changed analogously (Figure 4c,d). Differences in sun exposure (north vs. south aspect) did not result in significant changes in flesh color (Figure 5).

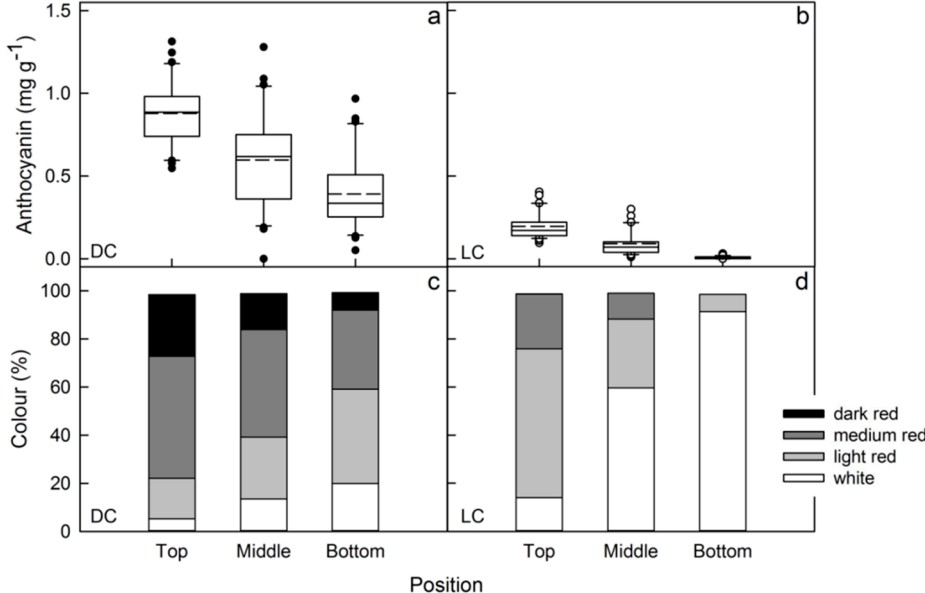

**Figure 4.** Effects of position along the central axis of a slender spindle tree on flesh anthocyanin content (**a**,**b**) and the color distribution (**c**,**d**) of red flesh apples of a dark-colored (DC) (**a**,**c**) and a light-colored (LC) clone (**b**,**d**).

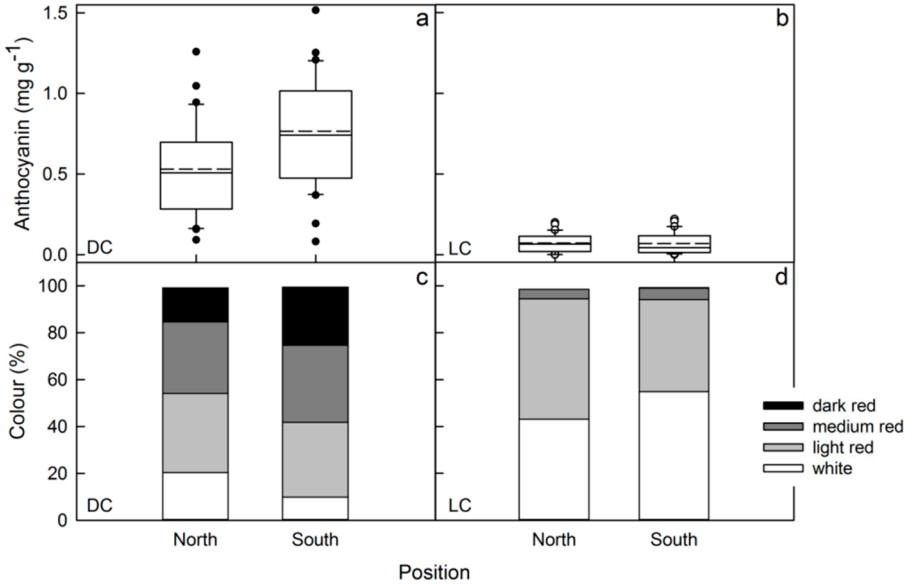

**Figure 5.** Effects of the exposure of the fruit to the sun on flesh anthocyanin content (**a**,**b**) and the color distribution (**c**,**d**) of red flesh apples of a dark-colored (DC) (**a**,**c**) and a light-colored (LC) clone (**b**,**d**).

The intensity of flesh color decreased when branches were shaded with hail net (Figure 6a,b). UV-exposure did not affect flesh color in the DC clone (Figure 7a,c) but in the LC clone flesh color decreased significantly when fruit were covered with UV-absorbing film (Figure 7b,d).

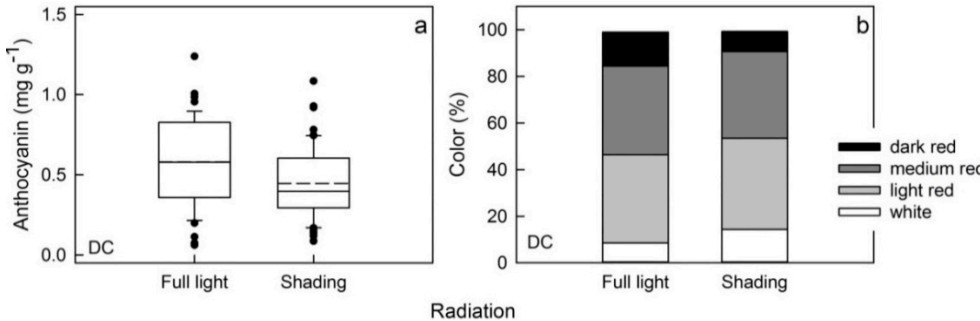

**Figure 6.** Effects of shading on the anthocyanin content (**a**) and the color distribution (**b**) of red flesh apples of a dark-colored (DC). Fruit exposed to full sunlight served as control. Shade reduced the incident photosynthetically active radiation by 94%.

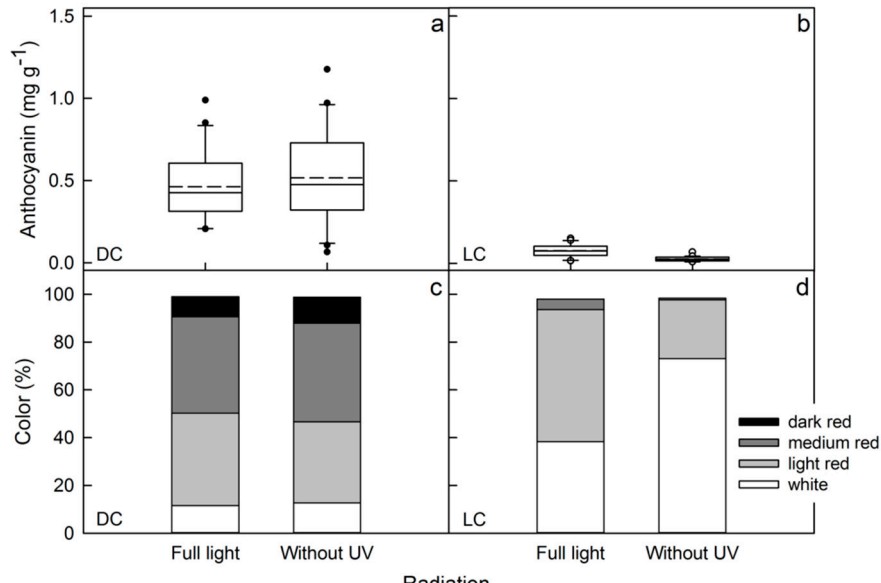

**Figure 7.** Effect of UV radiation on the anthocyanin content (**a,b**) and the color distribution (**c,d**) of red flesh apples of a dark-colored (DC) (**a,c**) and a light-colored (LC) clone (**b,d**). UV radiation was excluded by wrapping the fruit in a bag made of UV-absorbing film. The film absorbed 100% of the UV radiation. Fruit that was exposed to full sunlight served as control.

Compared with light fruit thinning, the medium and heavy fruit thinning both increased the red color in the flesh of both clones, DC and LC (Figure 8).

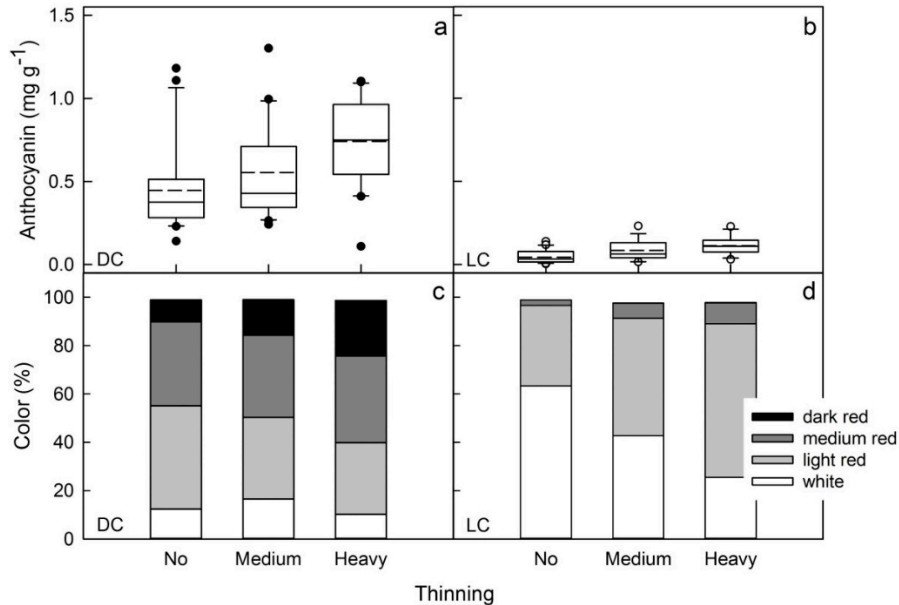

**Figure 8.** Effect of crop load on the anthocyanin content (**a**,**b**) and the color distribution (**c**,**d**) of red flesh apples of a dark-colored (DC) (**a**,**c**) and a light-colored (LC) clone (**b**,**d**). Crop load was varied by hand thinning to different levels. The mean numbers of fruit per tree were 200, 120 and 50 per tree (DC) and 120, 80 and 30 per tree (LC) for the light, medium and heavy fruit thinning treatments, respectively.

There was no consistent change in flesh color during storage of either clone (Figure 9a). In the DC clone, an increasing proportion of fruit developed severe flesh browning as storage time increased. There was no browning in the LC clone (Figure 9b,c).

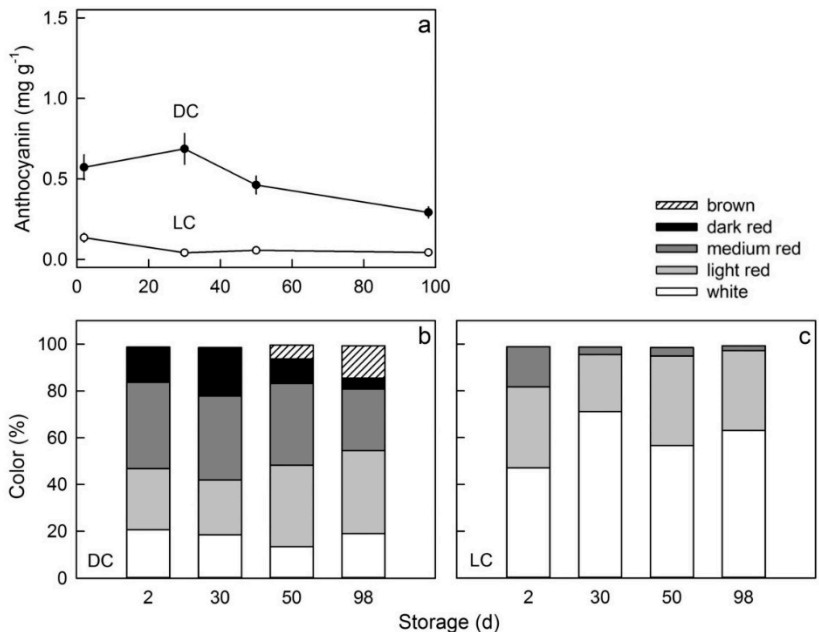

**Figure 9.** Time course of change in the anthocyanin content (**a**) and the color distribution (**b**,**c**) of red flesh apples of a dark-colored (DC) (**a**,**b**) and a light-colored (LC) clone (**a**,**c**) during storage. Fruit were held for up to 96 days after harvest.

## 4. Discussion

Our experiments established two important findings: (1) the method of estimating anthocyanin content using calibrated photographs, image analysis and a spectrophotometer, proved reliable and

sufficiently sensitive to quantify the effects of orchard management treatments imposed in field experiments, and (2) the effects of orchard management factors on anthocyanin biosynthesis were accounted for principally in terms of exposure to solar radiation.

*4.1. Quantifying Flesh Color Using Digital Photography and Image Analysis*

Measuring the redness of apple flesh based on digital photography and a calibration curve established between the L-value by a hand-held spectrophotometer and the anthocyanin absorption of flesh extract proved to be a quick, reliable and robust procedure for estimating flesh anthocyanin content. Compared to visual ratings, this method is more objective since it integrates spatial as well as color-intensity information, thereby producing a 'weighted' redness. This is difficult to achieve using visual ratings, particularly when comparing flesh color assessed by different operators. For the two clones investigated, the accuracy of the procedure was excellent when based on measurements of L-value. It accounted for more than 98% of the variation in anthocyanin content measured in vitro. Furthermore, the slope of the regression line of predicted vs. measured anthocyanin content of an independent data set was 1.12 which is a slight and consistent overestimation of the measured anthocyanin content ($r^2 = 0.98$).

Interestingly, predictions of anthocyanin content based on the a-values were less accurate. This result is counterintuitive since the relationship of red coloration and the a-value on a red/green axis should be superior. We suggest the inaccuracy was due to insufficient resolution with the high-intensity red colorations, where the absorption of the extracts increased markedly for fairly small changes of a-value.

*4.2. Light Stimulation of Anthocyanin Biosynthesis*

Light was the common factor explaining the response of anthocyanin biosynthesis to the orchard management factors investigated. This was demonstrated in the observation that red color increased for fruit in the top of the crown, without shade by foliage or by covering by UV-absorbent film. These observations are consistent with those by Honda et al. (2017) [14] on different clones of the same two red-fleshed categories. They reported decreased anthocyanin concentration as crop load increased, or for bagged vs. un-bagged fruit or for fruit grown in the shade vs. that exposed to the sun. Similarly, Matsumoto et al. (2018) [15] found decreased red color in the outer flesh as indexed by the a-value of the flesh when the fruit was shaded (by bagging), when crop load was high or when trees were defoliated. When bags were removed, color intensity increased again in the outer flesh. They observed no changes in the inner flesh [16]. Consistent with these reports was a decrease in anthocyanin content when shaded of red-flesh and white-flesh clones of crab apple (*Malus sylvestris*) [7].

*4.3. How Does Light Trigger Anthocyanin Biosynthesis in the Flesh?*

There is little transmission through apple skin and parenchyma [17]. Thus, the question arises how light triggers anthocyanin biosynthesis in the flesh of an apple when only the fruit surface is exposed to light. Two hypotheses are offered that deserve investigation. First, anthocyanin biosynthesis may be triggered by high concentrations of carbohydrates due to high rates of photosynthesis as a result of high light exposure of the canopy. This hypothesis is consistent with the effects of crop load, shading or defoliation observed in this and earlier studies [7,14,15].

Alternatively, photoreceptors may be considered. These perceive light levels at the fruit surface and then—via some mobile agent—trigger anthocyanin biosynthesis in the flesh. That light triggers anthocyanin biosynthesis is well established for the skin, for example in bagging experiments [14,16]. However, the flesh is not directly exposed. Blue light receptors (cryptochromes, phototropins) are potential candidates because radiation of 450 nm wavelength stimulates blush in apple skins [18]. Cryptochrome MdCRY2 is known to regulate anthocyanin biosynthesis [19]. In strawberries (*Fragaria × ananassa* Duch.), bio synthesis of anthocyanin is stimulated by 450 nm and by 730 nm wavelengths,

the absorption maxima of phototropins and phytochromes (Pfr) [20]. Clearly, at this stage, the evidence for a role of photoreceptors is not conclusive.

## 5. Conclusions

Digital photography plus image analysis is a quick and sufficiently precise method for quantifying redness of the flesh of apples because it is closely related to anthocyanin content as determined in the laboratory following extraction. The procedure is particularly useful because it does not require sophisticated instruments and a laboratory setting. Although we investigated only two representatives of the light- and dark-red flesh genotypes, we expect the relationship between anthocyanin absorption and the L-value to also hold for other red-fleshed genotypes.

The results revealed that light is the most important environmental factor regulating anthocyanin biosynthesis also in red flesh apple fruit. Increasing light exposure stimulated color development under orchard conditions. Light exposure can be maximized by maintaining an open canopy structure, pruning, thinning and by cultivating apples at sites with high levels of UV-B radiation.

**Author Contributions:** M.K. and E.G. designed the experiments; A.W. performed the experiments; A.W., E.G., M.K. analyzed the data; A.W. and E.G. prepared the draft of the paper; M.K. wrote the paper.

**Funding:** This research was funded in part by a grant from the IFORED group.

**Acknowledgments:** We thank Dirk Köpcke, Martin Brüggenwirth, Ulrich Hering, and Jennifer Kruse for their support in setting up the experiments and Sandy Lang and Bishnu Khanal for helpful comments on an earlier version of this manuscript.

**Conflicts of Interest:** The authors declare no conflict of interest.

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
