# Peer review of "Effect of Orchard Management Factors on Flesh Color of Two Red-Fleshed Apple Clones"

_horticulturae, doi:10.3390/horticulturae5030054_

Reviewer 1 Report

The method of determining the amount of anthocyanin using digital images is interesting.

In this paper, the image analysis is powerful method to objectively evaluate of coloring area in apple flesh.  Moreover, coloring is evaluated in 4 steps, which is relatively high precision. But, I think that this application was expected to be use in further experiments (for example, experiments using genes and experiments using multiple lines of apples).

Author Response

Reviewer 1

·         In this paper, the image analysis is powerful method to objectively evaluate of coloring area in apple flesh.  Moreover, coloring is evaluated in 4 steps, which is relatively high precision. But, I think that this application was expected to be use in further experiments (for example, experiments using genes and experiments using multiple lines of apples).

Thank you for the positive comment. Yes, we agree. The method has significant potential and will be applied in future research.

Reviewer 2 Report

Title - Should be more specific, what are the agronomic factors?

The quality of figure 1 is not satisfied. The scales overlap and are askew.

The experimental procesure can be presented using a flowchart.

Abstract - Should be more conclusive, what do you mean visible and UV light can stimulate anthocyanin synthesis in the flesh? Timimg? Levels? Which one is better?

The scale bars of LC in all figures are missing.

Author Response

Reviewer 2

·         Title - Should be more specific, what are the agronomic factors?

Accepted. We replaced “Agronomic Factors” by “Orchard Management Factors”. 

·         The quality of figure 1 is not satisfied. The scales overlap and are askew.

Accepted. We improved and replaced figure 1.

·         The experimental procedure can be presented using a flowchart.

We do not share the referee’s view point. In order to allow the method to be used also by fellow researchers, we need to provide a detailed description of the method and the materials used throughout the experiments. This information cannot be provided in a flow chart only.

·         Abstract - Should be more conclusive, what do you mean visible and UV light can stimulate anthocyanin synthesis in the flesh? Timimg? Levels? Which one is better?

Accepted. We now specify light quality (shading, UV experiments). The concluding statement at the end of the abstract was reworded.

·         The scale bars of LC in all figures are missing.

Rejected. The error bars are included, but smaller than data symbols (see also statement in lines 126/127).

Reviewer 3 Report

Comments for the Author

In this manuscript, the authors established a digital-image analysis method to quantify the red-coloration of apple flesh which can reflect the anthocyanin content of fruits. In addition, by utilizing this analysis method, the accumulation patterns of anthocyanin and the redness of fruits from two apple cultivars were tested with or under varied agronomic parameters or conditions, such as position, light shading, UV exposure, trimming and storage time. Based on the correlation analysis of the anthocyanin contents and the different factors, it is identified that the light exposure is the most prominent factor that regulates the flesh coloration for red apple fruits.

Since the orchard conditions and managements responsible for the color formation of fruit flesh are crucial for the improvement of apple quality, this manuscript provided valuable method and knowledge for understanding and performing appropriate management for the two apple cultivars and indications for other apple genotypes or even other fruit plants. If the authors could address the issues below, the manuscript would meet the high standard for publication.

Main issues:

It would be better to separate the figures for position factors each into two figures as the position in the canopy and the exposure relative to the sun respectively.

Please indicate the temperature range of the storage rather than just described as “below 3 °C”.

Please describe the “homogenous, standardized illumination” in more detail and the exposure parameters for the photographs.

Please pay attention to the use of abbreviations. It is recommended to give the full name when first time an abbreviation appears then use the abbreviation in the rest part. For example, the “DAFB” for “days after full bloom”.

In the Experiments section, it would be better to illustrate as “To test the relationship between the flesh color and…” or “To test the influence of…on the flesh color”, rather than “Flesh color was recorded as a function of…”.

Some other issues:

Line 27, “In these genotypes, the red color…”

Line 31, “… and dark-colored.”

Line 38, “… flowers or fruits.”

Line 42, “The concentration of anthocyanin in the flesh and hence the color intensity,…”

Line 45, “…, and hence the skin and flesh color of fruit;…”

Line 77, “… the equatorial plane of DC and LC fruits.”

Author Response

Reviewer 3

Main issues:

·         It would be better to separate the figures for position factors each into two figures as the position in the canopy and the exposure relative to the sun respectively.

Accepted. We separated Figure 4 into two separate Figures (Figure 4 and 5), all subsequent figures were renumbered accordingly.

·         Please indicate the temperature range of the storage rather than just described as “below 3 °C”.

Accepted. We are now more specific. The temperature averaged 2 ±1°C.

·         Please describe the “homogenous, standardized illumination” in more detail and the exposure parameters for the photographs.

Accepted. We specified the illumination and provide details on number of tubes, watts and kelvin. 

·         Please pay attention to the use of abbreviations. It is recommended to give the full name when first time an abbreviation appears then use the abbreviation in the rest part. For example, the “DAFB” for “days after full bloom”.

Rejected. Please check. The abbreviation is defined at first mention (see abstract line 16).

·         In the Experiments section, it would be better to illustrate as “To test the relationship between the flesh color and…” or “To test the influence of…on the flesh color”, rather than “Flesh color was recorded as a function of…”.

We do not agree with the referee here. We describe the individual experiments in individual paragraphs. The first sentence of anyone paragraph is an introductory statement indicating the purpose of the particular experiment.

Some other issues:

All proposed changes accepted.